# Remimazolam Induction in a Patient with Super-Super Obesity and Obstructive Sleep Apnea: A Case Report

**DOI:** 10.3390/medicina59071247

**Published:** 2023-07-05

**Authors:** Sou Hyun Lee, Hyeji Han

**Affiliations:** Department of Anesthesiology and Pain Medicine, Dongsan Medical Center, Keimyung University School of Medicine, Daegu 42601, Republic of Korea

**Keywords:** anesthesia, case report, morbid obesity, remimazolam

## Abstract

*Background*: With the rising prevalence of obesity, anesthesiologists are expected to increasingly encounter patients with obesity, which poses challenges for anesthetic management. The use of remimazolam, an intravenous anesthetic agent approved in 2020, may be beneficial in these patients. However, its use in patients with super-super obesity remains underexplored. *Case Description*: A 55-year-old woman with a body mass index (BMI) of 60.6 kg/m^2^ and moderate obstructive sleep apnea (OSA) underwent laparoscopic sleeve gastrectomy under general anesthesia. The transnasal humidified rapid-insufflation ventilatory exchange (THRIVE) technique was used along with the administration of remimazolam at a rate of 6 mg/kg/h based on the total body weight. The patient was sedated within 125 s without any signs of hemodynamic instability, and the surgery was completed successfully. *Conclusions*: This case study demonstrates the potential effectiveness of remimazolam infusion for inducing general anesthesia in patients with super-super obesity. The infusion rate, derived from the total body weight, yielded an outcome comparable with that observed in individuals without obesity. Further studies with larger cohorts are required to confirm these findings.

## 1. Introduction

Given the increasing prevalence of obesity worldwide, anesthesiologists are more likely to encounter patients with obesity in operating rooms. However, anesthetic management of patients with obesity presents several challenges, such as desaturation due to the presence of excessive truncal fat that displaces the diaphragm cephalically, which leads to a reduction in the total lung capacity and functional residual capacity, as well as an increase in oxygen consumption [1,2,3]. Approximately 40–90% of patients with obesity are also affected by obstructive sleep apnea (OSA) [4]. OSA is characterized by the persistent narrowing and subsequent closure of the pharynx during sleep, an occurrence that increases the likelihood of upper airway collapse, which can subsequently result in hypoxia [1]. In addition, hemodynamic instability may occur due to an increase in the volume of distribution and the need for the administration of higher doses of sedatives [5,6].

Remimazolam, an intravenous anesthetic agent approved in 2020 [7], is an alternative to propofol, which is the most widely used anesthetic agent in patients with obesity [8]. Remimazolam is structurally similar to midazolam; however, it is characterized by more rapid onset and offset [9], lesser respiratory depression compared with that of propofol, and more stable hemodynamics [10], making it a potential alternative for patients with obesity. This case study focuses on the application of remimazolam for the induction of general anesthesia in a patient with super-super obesity. As its efficacy in patients with super-super obesity remains unexplored, we present a case wherein remimazolam successfully induced general anesthesia in a patient with super-super obesity to demonstrate its potential safety.

## 2. Detailed Case Description

This manuscript adhered to the CARE guidelines [11]. The patient was a 55-year-old woman (height, 152 cm; weight, 140 kg) with a BMI of 60.6 kg/m^2^ who had moderate OSA. OSA was managed with continuous positive airway pressure therapy during sleep. The patient also had asthma and depression. As the patient had not been able to lose weight through diet and exercise, she was scheduled to undergo laparoscopic sleeve gastrectomy under general anesthesia. Preoperative airway evaluation revealed a modified Mallampati class IV airway with a full range of neck motion. Pulmonary function tests revealed mild restrictive airway obstruction (forced expiratory volume in 1 s (FEV_1_), 53% of the predicted value; forced vital capacity (FVC), 52% of the predicted value; and FEV_1_/FVC, 77% of the predicted value). Preoperative chest radiography and computed tomography (CT) revealed cardiomegaly. However, transthoracic echocardiography demonstrated normal left ventricular function without ventricular hypertrophy.

The patient was placed in a ramped position on arrival at the operating room. Monitoring devices, including a pulse oximeter and a three-lead electrocardiogram, were attached. Depth of anesthesia was assessed with the patient state index (PSI) using SedLine^®^ (Masimo Corporation, Irvine, CA, USA). A neuromuscular monitor was attached to the right hand of the patient, and a 20G radial artery catheter was inserted into the left radial artery for continuous blood pressure monitoring.

Transnasal high-flow oxygen therapy (transnasal humidified rapid-insufflation ventilatory exchange (THRIVE); Optiflow™; Fisher & Paykel, Auckland, New Zealand) was initiated at a rate of 30 L/min and fractional inspired oxygen of 1.0 for 2 min. Pre-induction arterial blood gas analysis (ABGA) revealed a partial pressure of oxygen (PaO_2_) of 382 mmHg, partial pressure of carbon dioxide (PaCO_2_) of 49 mmHg, and arterial oxygen saturation (SatO_2_) of 100% (Table 1). Remimazolam infusion (6 mg/kg/h, total body weight) was initiated using the Agilia^®^ Connect Infusion System. Loss of consciousness (LoC), which was defined as the patient not responding to being tapped on the shoulder [12], occurred 125 s after initiating remimazolam infusion. The THRIVE flow rate was increased to 70 L/min, followed by the administration of remifentanil and rocuronium. Intubation was performed 300 s after initiating remimazolam infusion. Mechanical ventilation was initiated thereafter, with an immediate post-apneic end-tidal CO_2_ level of 42 mmHg. Anesthesia was maintained with sevoflurane inhalation and remifentanil infusion. Peripheral oxygen saturation briefly decreased to 88% during intubation; however, it rapidly recovered to 99% upon the initiation of mechanical ventilation. A second ABGA, performed 3 min post-intubation, revealed a PaO_2_ of 127 mmHg, PaCO_2_ of 49.5 mmHg, and SatO_2_ of 98.9%. Figure 1 presents the vital signs and changes in the PSI during anesthesia induction and intubation.

The pre-induction blood pressure and heart rate of the patient were 154/87 mmHg and 86 beats/minute, respectively. The lowest blood pressure prior to intubation was 138/99 mmHg, whereas the heart rate was maintained within the range of 86–93 beats/minute. Vasopressors and inotropic agents were not administered during the induction period due to the risk of developing hypotension (systolic blood pressure < 80 mmHg) or bradycardia (heart rate < 60 beats/minute).

The duration of surgery was 153 min, and 400 mg of sugammadex was administered for neuromuscular reversal. The patient was subsequently transferred to the post-anesthetic care unit. She did not experience intraoperative awareness and was transferred to the general ward 30 min later. No complications were observed at the time of discharge on the fifth postoperative day.

## 3. Discussion

The THRIVE technique was utilized for apneic oxygenation between induction and intubation in the present case. The patient was successfully sedated within 125 s using remimazolam infusion at a rate of 6 mg/kg/h, with a total dosage of 30.3 mg (0.22 mg/kg). No signs of hemodynamic instability were observed. Desaturation occurred 300 s after induction, following the initiation of the THRIVE technique. The PSI score at the time of LoC was 81.

As various clinical studies have been performed using continuous infusion [13,14,15], the current medication guide recommends the administration of remimazolam via infusion to induce general anesthesia [16]. General anesthesia can be induced at a rate of 6 mg/kg/h or 12 mg/kg/h. In contrast to bolus induction [17], an increase in the infusion rate does not lead to a proportional decrease in blood pressure [15]. However, Kim et al. [18] reported that higher infusion rates may trigger non-IgE-mediated anaphylactic responses (occurring at a rate of 0.26% with 12 mg/kg/h compared with 0.08% with 6 mg/kg/h) to dextran, a stabilizer present in remimazolam. Therefore, induction of anesthesia at a rate of 6 mg/kg/h is considered more suitable. In the present case, the patient did not develop any anaphylactic symptoms (hypotension or skin erythema) following the administration of 30.3 mg of remimazolam for 125 s.

Midazolam, which is structurally similar to remimazolam, is a benzodiazepine that produces lesser respiratory depression and hemodynamic changes. However, it has the disadvantages of slower onset (5–10 min) and recovery (5–20 min) compared with that of remimazolam. The slower onset and recovery of midazolam may be attributed to its metabolism in the liver, which leads to the production of an active substance. In contrast, remimazolam has a more rapid onset (1–4 min) and recovery (40 min), and it is metabolized to the inactive metabolite CNS7054 by nonspecific tissue esterase [19,20,21]. These properties make the use of remimazolam particularly advantageous in patients with obesity. This is further supported by the fact that the context sensitive half-time of midazolam and remimazolam, that is, the time taken for the concentration of these drugs to decrease by 50% in the blood after they are continuously administered for 4 h and then discontinued, are 60 and 6.8 min, respectively [13,22].

Unlike the infusion of propofol and midazolam, which are known to reduce blood pressure in a dose-dependent manner [23,24], continuous infusion of remimazolam demonstrated no dose-cumulative effect on blood pressure in the present case. This observation aligns with the findings of previous studies involving patients without obesity receiving remimazolam infusion, where changes in blood pressure and heart rate were maintained within 20% of the baseline values during anesthesia induction [25,26].

LoC is typically assessed using the modified observer’s assessment of alertness/sedation scale; however, this scale is a subjective indicator with a potential for assessor bias. Representative objective measures for evaluating consciousness in the field of anesthesia include the patient state index (PSI), which was used in the present study, and the bispectral index (BIS). PSI and BSI express the depth of anesthesia by analyzing electroencephalograms. An adequate depth of anesthesia is typically observed at a PSI of 25–50 and a BIS of 40–60. However, a target BIS of 70 is suggested due to the high power of the beta waves (13–15 Hz) with the use of remimazolam [13,27]. Studies monitoring the subjective indicators of LoC and PSI have reported that the PSI remained above 60 for 5 min following the administration of remimazolam [28]. Similarly, in the present case, the PSI remained above 60 for the first 4 min after the administration of remimazolam, which is consistent with the findings of previous studies. However, it is important not to rely solely on objective indices to evaluate a patient’s LoC. This is due to the time delay that exists between the onset of unconsciousness and the manifestation of changes in EEG-based indices, along with the significant variability and overlap between LoC and non-LoC states [29]. Therefore, it is crucial to verify the LoC using not only objective indicators but also subjective ones.

Continuous infusion is associated with some drawbacks, such as the requirement of complex preparation for anesthesia and the risk of overdose. Consequently, there has been a rise in the number of studies exploring the use of alternatives, such as single-bolus administration. A dosage of 0.19 mg/kg is currently recommended for inducing anesthesia using single-bolus administration in patients aged 60 years or older [28]. It must be noted that, in contrast to continuous infusion, this method can induce more pronounced hemodynamic shifts if the dosage is increased, necessitating its careful application [30]. Thus, until its safety is proven, it would be advisable to avoid the use of single-bolus administration in patients with obesity, as a high body weight could result in an excessive dose of remimazolam being administered concurrently.

The average duration to achieve LoC using remimazolam infusion at 6 mg/kg/h in patients without obesity, according to the total body weight, ranges between 97.2 and 102 s [13,19]. Thus, the duration of LoC observed in the present case (125 s) was comparable with those of prior studies. Moreover, the PSI at LoC was 81 (below 85), which is consistent with the value observed in a previous study by Chae et al. [28] that used the same definition of LoC. Collectively, these results advocate for the calibration of the initial dosage to be based on the total body weight in patients with obesity. In addition, considering the structural similarities between remimazolam and midazolam [9] and given that total body weight-based dosing has already been implemented for midazolam induction in patients with obesity [31], it is reasonable to suggest that the same approach should be used for determining the dosage of remimazolam for anesthesia induction in patients with obesity.

The successful administration of remimazolam in the present case demonstrated its efficacy in inducing anesthesia in patients with a high BMI. The patient was sedated within 125 s without any signs of hemodynamic instability. The observed PSI scores were consistent with those reported in previous studies, further validating our approach. Despite the potential of remimazolam to trigger anaphylactic responses, no such symptoms were observed, indicating that the selected infusion rate was safe and effective. However, these findings are based on a single patient and may not be generalizable to all patients. Therefore, further studies involving a larger population of individuals with obesity are required.

There are some limitations to our study. A major limitation was that the lack of upper airway patency maintenance during anesthetic induction precluded the examination of the impact of BMI on apnea duration. In addition, we did not measure the immediate post-apneic PaCO_2_ level, thereby making it impossible to observe any potential PaCO_2_ increases during the induction period. Furthermore, our investigation was solely focused on the use of remimazolam, and any combination of opioids or other adjuncts were excluded. Given the typical multidrug approach in anesthesia induction, our findings may not be applicable to all clinical scenarios where remimazolam is used for inducing anesthesia in patients with obesity.

## 4. Conclusions

Given that an ideal anesthetic is frequently characterized by its minimal hemodynamic and respiratory depressant effects [32], this case report describes the effective application of remimazolam for the induction of general anesthesia in a patient with super-super obesity. An infusion rate of 6 mg/kg/h, derived from the total body weight, yielded an outcome comparable to that observed in individuals without obesity.

## Figures and Tables

**Figure 1 medicina-59-01247-f001:**
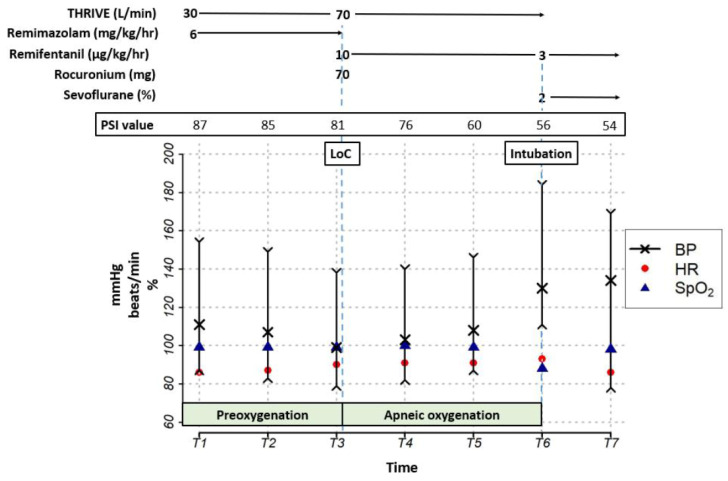
Anesthetic record during induction. THRIVE, transnasal high-flow oxygen therapy; PSI, patient state index; LoC, loss of consciousness; BP, blood pressure; HR, heart rate; SpO_2_, peripheral oxygen saturation; T1, preinduction; T2, 1 min after induction; T3, 2 min after induction; T4, 3 min after induction; T5, 4 min after induction; T6, 5 min after induction; T7, 3 min after intubation.

**Table 1 medicina-59-01247-t001:** Arterial blood gas analysis results.

	Pre-Induction ABGA	3 min Post-Intubation ABGA
pH (mEq/L)	7.384	7.371
PaCO_2_ (mmHg)	49	49.5
PaO_2_ (mmHg)	382	127
HCO_3_^−^ (mmol/L)	29.3	28.7
BE (mmol/L)	3.3	2.6
SatO_2_ (%)	100	98.9
Hb (g/dL)	12.4	12.1
Hct (%)	38.1	37.2
Lactic acid (mmol/L)	1.6	1.2

ABGA, arterial blood gas analysis; PaCO_2_, partial pressure of carbon dioxide; PaO_2_, partial pressure of oxygen; HCO_3_^−^, bicarbonate; BE, base excess; SatO_2_, arterial oxygen saturation; Hb, hemoglobin; Hct, hematocrit.

## Data Availability

The datasets generated and/or analyzed in the current study are available from the corresponding author upon reasonable request.

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
