# Peer review of "Remimazolam Induction in a Patient with Super-Super Obesity and Obstructive Sleep Apnea: A Case Report"

_medicina, 2023, doi:10.3390/medicina59071247_

Round 1

Reviewer 1 Report

This case report submitted by Lee and Han describes the use of remimazolam in the anesthetic management of an extremely obese individual with obstructive sleep apnea. Although the subject matter in the case is of modest interest to an anesthesiologist, it is excessively long and contains superfluous information of little relevance to the use of remimazolam in this clinical circumstance.

The following recommendations are made:

1.     Introduction

a.     1st 2 sentences can be omitted.

b.     Shorten last paragraph of introduction

2.     Case Report

a.     Omit 1st sentence.

b.     Omit pictures of upper airway, chest radiograph and CT scan; Can just state she had cardiomegaly on these studies;

c.     Shorten anesthetic description on lines 72 through 97.

d.     Table 1 can be omitted. You can just state that a subsequent blood gas showed no hypoxemia and no change in PaCO2.

3.     Discussion

a.     The authors purport that this is a case report describing the use of remimazolam. Therefore, there is little utility in the extensive description of the THRIVE technique on lines 143-161. You can merely state that the THRIVE  technique was used to apneic oxygenation between induction and intubation.

b.     Line 177-178: Given the response time of a PtcCO2 electrode, it seems unrealistic to use it as a measure to track changes in PaCO2 during induction. It is suggested that this sentence be eliminated.

Author Response

"Dear Reviewer,

Thank you for taking the time to review my manuscript. Your comments have been invaluable in enhancing the quality of the manuscript. As suggested, I have significantly trimmed the sections pertaining to THRIVE and have instead incorporated a comprehensive literature on remimazolam, along with comparisons to the findings from this case. Attached below are my responses to the 'reviewer report' and the completed CARE checklist. I trust these changes improve the clarity and contribution of my research.

Once again, I appreciate your thoughtful and constructive feedback.

Kind regards, Sou Hyun Lee, MD, PhD

Reviewer 2 Report

Remimazolam Induction in a Patient with Super-Super Obesity and Obstructive Sleep Apnea: A Case Report

This manuscript aimed to describe the potential effectiveness of remimazolam infusion for inducing general anesthesia in patients with super-super obesity. Overall, this topic is important and the results are promising.

However, some concerns appeared after reading the whole manuscript.

1. CARE guidelines need reference, such as,

Riley DS, Barber MS, Kienle GS, Aronson JK, von Schoen-Angerer T, Tugwell P, Kiene H, Helfand M, Altman DG, Sox H, Werthmann PG, Moher D, Rison RA, Shamseer L, Koch CA, Sun GH, Hanaway P, Sudak NL, Kaszkin-Bettag M, Carpenter JE, Gagnier JJ. CARE guidelines for case reports: explanation and elaboration document. J Clin Epidemiol. 2017 Sep;89:218-235. doi: 10.1016/j.jclinepi.2017.04.026.

2. More space need to be placed between “Pre-induction ABGAand 3 minutes post-intubationABGA.

3. Thus, the duration of LoC observed in this case (125 s) was not statistically different from 134

those of prior studies.Because the current study did not conduct statistical inference, thus, the statistically in this sentence might be not appropriate.

 Minor editing of English language required.

Author Response

Dear Reviewer,

Thank you for dedicating your valuable time to reviewing my manuscript. Your feedback has been instrumental in helping me refine and improve my work. I'm particularly grateful for your provision of the correct reference style for the CARE guidelines, which was an oversight in the original draft. Please find attached my responses to the 'reviewer report'.

Once again, I sincerely appreciate your interest and invaluable input on my manuscript.

Kind regards, Sou Hyun Lee, MD, PhD

Round 2

Reviewer 1 Report

The authors have adequately responded to my comments.